# Clumped isotope evidence for Early Jurassic extreme polar warmth and high climate sensitivity

Thomas Letulle[1], Guillaume Suan[1], Mathieu Daëron[2], Mikhail Rogov[3], Christophe Lécuyer[1], Arnauld Vinçon-Laugier[1], Bruno Reynard[1], Gilles Montagnac[1], Oleg Lutikov[3], Jan Schlögl[4].

[1] Univ Lyon, UCBL, ENSL, UJM, CNRS, LGL-TPE, F-69622, Villeurbanne, France

[2] Laboratoire des Sciences du Climat et de l'Environnement, LSCE/IPSL, CEA-CNRS-UVSQ, Université Paris-Saclay, Orme des Merisiers, F-91191 Gif-sur-Yvette Cedex, France

[3] Geological Institute of Russian Academy of Sciences, Laboratory of Phanerozoic Stratigraphy.

[4] Comenius University, Faculty of Natural Sciences, Department of Geology and Palaeontology, Mlynská dolina G, 842 15 Bratislava, Slovak Republik

*Correspondence to*: Thomas Letulle (thomas.letulle@univ-lyon1.fr)

**Abstract.** Periods of high atmospheric $CO_2$ levels during the Cretaceous-Early Paleogene (~140 to 34 My ago) were marked by very high polar temperatures and reduced latitudinal gradients relative to the Holocene. These features represent a challenge for most climate models, implying either higher-than-predicted climate sensitivity to atmospheric $CO_2$, or systematic biases or misinterpretations in proxy data. Here, we present a reconstruction of marine temperatures at polar (>80°) and mid (~40°) paleolatitudes during the Early Jurassic (~180 My ago) based on the clumped isotope ($\Delta_{47}$) and oxygen-isotope ($\delta^{18}O_c$) analyses of shallow buried pristine mollusc shells. Reconstructed calcification temperatures range from ~8 to ~18°C in the Toarcian Arctic and from ~24 to ~28°C in Pliensbachian mid-paleolatitudes. These polar temperatures were ~10-20°C higher than present along with reduced latitudinal gradients. Reconstructed seawater oxygen isotope values ($\delta^{18}O_w$) of $-1.5$ to $0.5$‰ VSMOW and of $-5$ to $-2.5$‰ VSMOW at mid and polar paleolatitudes, respectively, point to a significant freshwater contribution in Arctic regions. These data highlight the risk of assuming the same $\delta^{18}O_{sw}$ value for $\delta^{18}O$-derived temperature from different oceanic regions. These findings provide critical new constraints for model simulations of Jurassic temperatures and $\delta^{18}O_{sw}$ values and suggest that high climate sensitivity is a hallmark of greenhouse climates since at least 180 My.

## 1 Introduction

Proxy data indicate that the Cretaceous-Early Paleogene (~140 to 34 My ago) was characterized by high atmospheric $CO_2$ concentrations, extreme polar warmth and reduced latitudinal temperature gradients (Sluijs et al., 2006; Suan et al., 2017; Evans et al., 2018). Most state-of-the-art climate models hardly reproduce such features, implying either a higher climate sensitivity under greenhouse conditions or systematic biases in proxy data interpretation (Huber and Caballero, 2011; Laugié et al., 2020; Zhu et al., 2020). It remains unclear whether higher climate sensitivity is unique to the Cretaceous-Early Paleogene world or is rather a hallmark of Earth's climate under high atmospheric $pCO_2$. Temperature proxies sensitive to overwriting under important burial, such as molecular or clumped-isotope thermometry (Henkes et al., 2014; Fernandez et al., 2021; Hemingway and Henkes, 2021), have seldom been applied to older sediments owing to their generally higher thermal maturity

(Robinson et al., 2017; Ruebsam et al., 2020; Fernandez et al., 2021). Consequently, current temperature estimates predating the Cretaceous period are mostly derived from the oxygen isotope composition of marine carbonate fossils ($\delta^{18}O_c$), with well-known limitation related to uncertainties in the past $\delta^{18}O$ signature of seawater ($\delta^{18}O_w$) (Epstein et al., 1953; Roche et al., 2006; Laugié et al., 2020).

Here, we use carbonate clumped isotope thermometry ($\Delta_{47}$), to simultaneously constrain the calcification temperatures and associated $\delta^{18}O_w$ values of marine carbonate shells (mostly aragonite) collected from Lower Jurassic sedimentary successions with exceptionally shallow to moderate burial depths spanning subtropical to polar paleolatitudes. We compare our results to existing Jurassic to Eocene climate proxy data and simulations and discuss their implications for climate sensitivity under greenhouse conditions.

## 2 Geological settings

### 2.1 Polovinnaya River

The Polovinnaya River section is located in northern Siberia (72°36'05" N, 107°58'52.2" E), and was located near the north pole during the Early Jurassic (Fig. 1). Our bivalve samples come from between 0 and 14m in the section and belong to the Toarcian (Suan et al., 2011). This interval consists of silty shale slightly enriched in organic carbon (TOC ~0.5%). Except for localized carbonate concretions, there is no carbonate fraction in the sediment. The studied interval has been previously correlated to the lower Toarcian Serpentinum ammonite zone based on biostratigraphy of foraminifera and dinoflagellate cyst, and lithostratigraphic correlation with other sections of the basin (Suan et al., 2011). This section records very abundant *Dacryomya* bivalve shells (Fig. S1), an opportunistic suspension-feeder genus tolerant to poorly oxygenated waters, which preferred conditions with weak hydrodynamics (Zakharov and Shurygin, 1978). Few belemnite rostra were also recorded as well as isolated fish scales and teeth. Overall the fossil assemblage indicates fully marine conditions within proximity of the continents as evidenced by the occurrence of wood debris. The section has undergone low burial as suggested by the low values of Rock-Eval Pyrolysis $T_{max}$ (mean = 420°C) previously measured in the host sediments (Suan et al., 2011). Regional stratigraphy from the more distal Anabar area suggests local overburden not exceeding 1000 m: a total overburden (Lower Toarcian to Valanginian) of about 380 m is recorded in the Anabar River area (Nikitenko et al., 2013) located 200 km East of the Polovinnaya section, which may be extended to about 1000 m when adding Valanginian-Cenomanian overburden from the more distal Bol'shoi Begichev islands. Modern local geothermal fluxes are lower than 50 mW/m$^2$ (Kerimov et al., 2020), indicative of a low geothermal gradient (<25 °C/km). Assuming a warm mean surface temperature of 10°C, the 1000 m of overburden and assuming that the geothermal gradient of the Siberian craton did not significantly change in the last 200 My, maximum burial temperatures of around 35°C can be estimated for the studied specimens from Polovinnaya river.

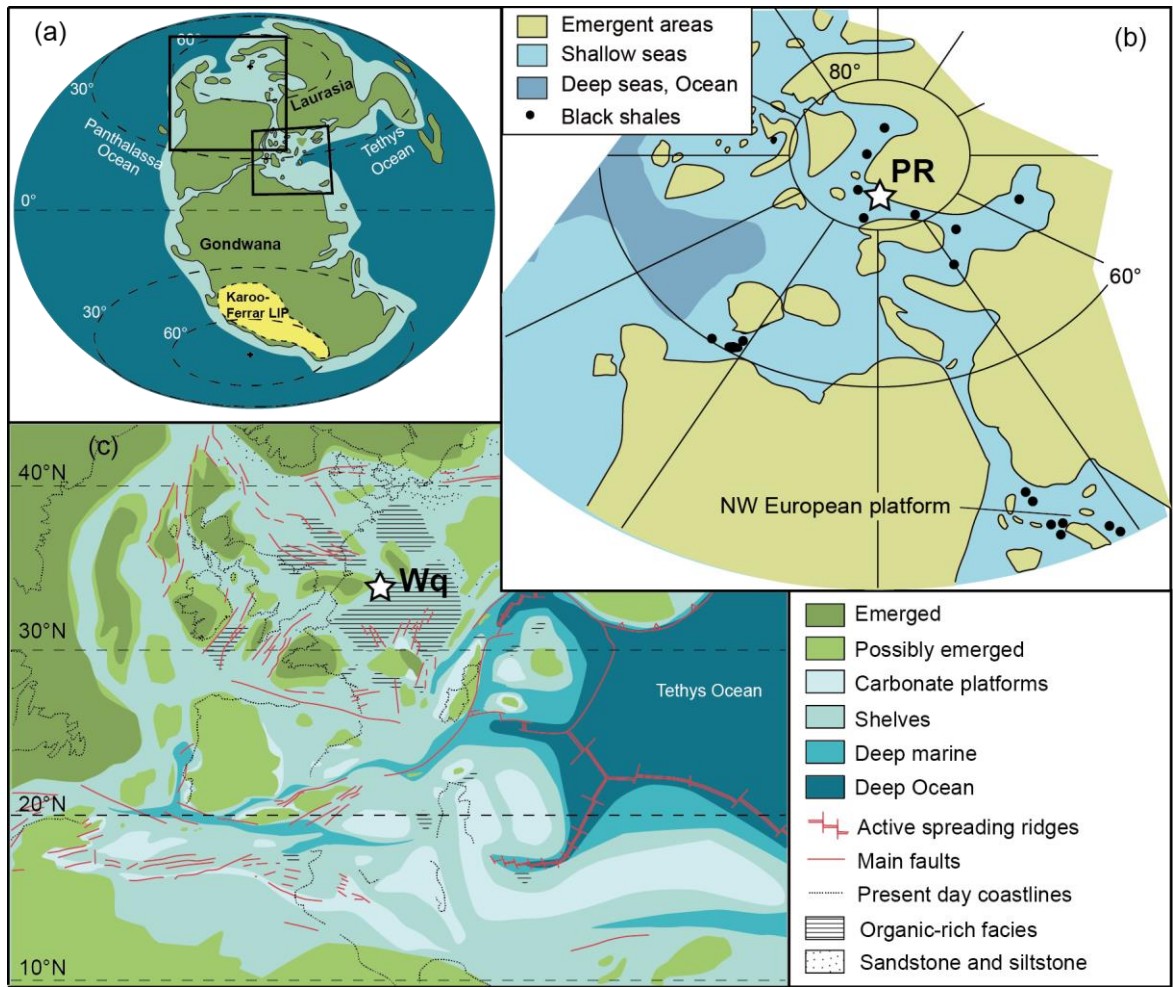

**Figure 1. Location of the studied sites with regard to Toarcian (Early Jurassic) geography.** (a) Global map modified from Dera et al. (2009). (b) Arctic map modified from Nikitenko and Mickey (2004). (c) Tethyan map modified from Thierry, (2000). Localities: PR = Polovinnaya River; Wq = Warcq.

### 2.2 Warcq

Samples from north-eastern Paris Basin were collected in 2014 from a temporary road cutting located near Warcq, Ardennes, France (49°45'21.6"N, 4°39'28.8"E). They consist of grey silty claystone with lenses of packed carbonated shell fragments, mainly from a variety of bivalves (*Grammatodon, Malletia, Limea, Oxytoma*) and few ammonoids (*Beaniceras, Aegoceras?,* Dactylioceratidae) (Fig. 2). The lithology, fossil preservation and assemblages of the sampled beds are similar to those described by Thuy et al. (2011), from a

nearby site of Sedan and dated from the Pliensbachian Davoei zone. The sampled levels are therefore tentatively attributed to the lower Pliensbachian Davoei ammonite zone. Mean $T_{max}$ values of 425°C and maximum burial temperatures near 60°C have been reported for Pliensbachian sediments from NE Paris Basin boreholes where the Davoei zone bears at ~1100 m deep in the EST 433 borehole, some 150 km south from Warcq (Blaise et al., 2014), and lies at ~860 m in the Montcornet borehole, some 50 km west from Warcq (Disnar et al., 1996;

Bougeault et al., 2017). These burial temperatures and depth should be regarded as an upper limit, as the very proximal sampling area near Warcq was repeatedly emerged during the Mesozoic and hence shows a much-thinner Mesozoic cover of ~500 m than these more distal sites (Waterlot et al., 1960). Assuming 860 m of overburden, ~300m of Cretaceous overburden eroded during the Cenozoic based on Paris basin thermal history

(Brigaud et al., 2020), a Mesozoic surface temperature of ~20 °C and canonical continental geothermal gradient of ~35 °C/km, maximal burial temperature of ~60 °C can be estimated for the studied specimens from Warcq.

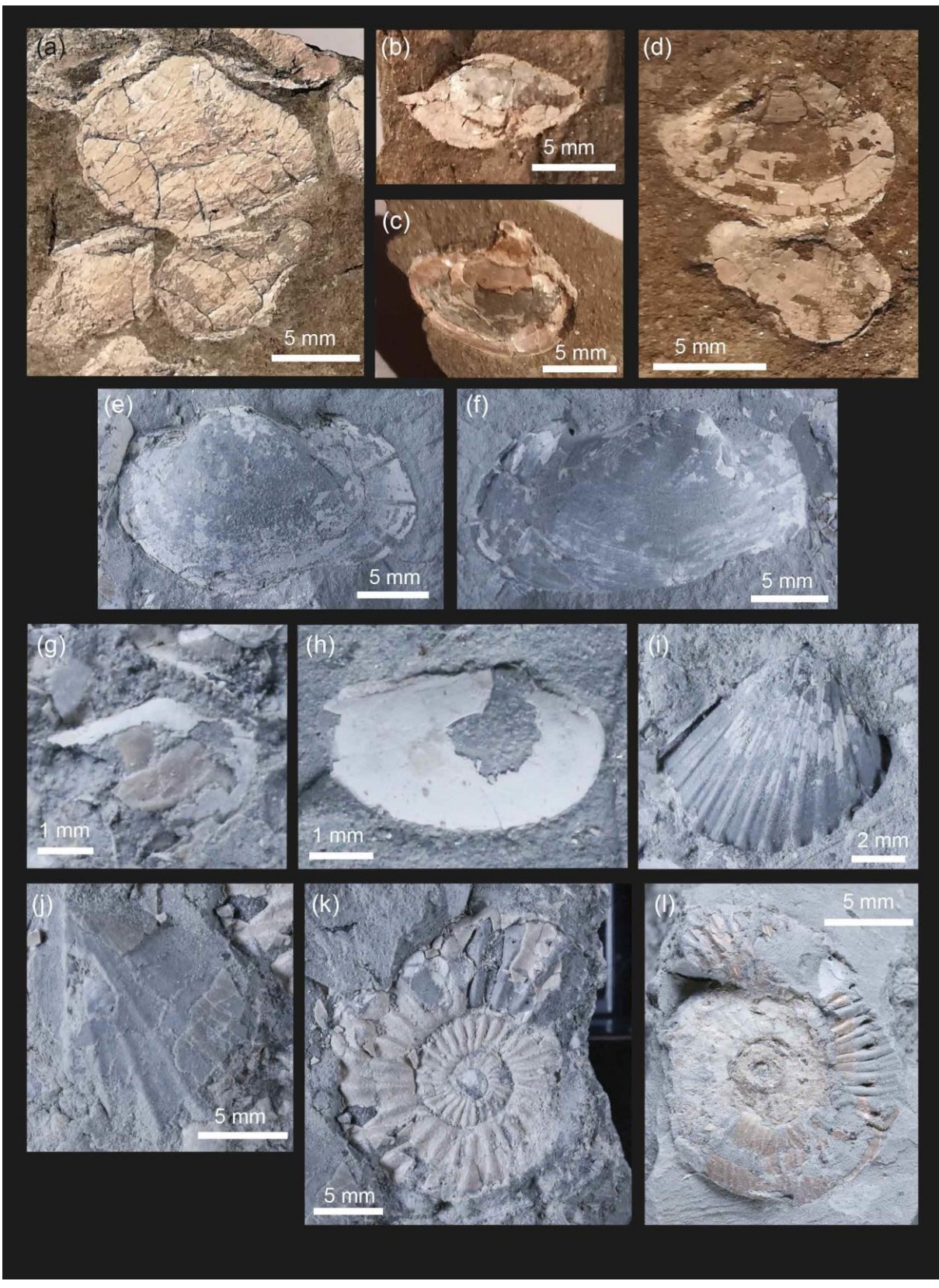

**Figure 2. Selected specimens from the sampled successions.** (a-d). Polovinnaya River section (Toarcian), (e-l). Warcq section (Pliensbachian); (a). *Dacryomya jacutica*, specimen Pol 29 (on the surface of a carbonate concretion); (b). *Dacryomya jacutica*, specimen Pol 13; (c-d). *Dacryomya jacutica*, specimen Pol 5; (e-f). *Grammatodon* sp., specimen ARD-01(inner mould after sampling of the shell); (g-h). *Malletia* sp.; (i). *Limea* sp., specimen ARD-03 (inner mould after sampling of the shell); (j). *Oxytoma* sp.ind. (inner mould with remains of a calcite shell) (k). *Aegoceras ?*, specimen ARD 06; (l). Dactylioceratidae indet., specimen ARD-07.

## 3 Material and Methods

### 3.1 Sampled material

The two studied sites present exceptionally rare records of aragonite preservation for the Lower Jurassic interval. *Dacryomya* shells are the most abundant macrofossil and the only bivalve genus to occur in the Polovinnaya River section. They are very abundant in the lower part of the section (0 to 8 m). They are mainly represented by adult shells, while juveniles are common in only few levels. They appear as ~1 cm distinct individual or detached valves, sometimes close to each other (Fig. 2). The carbonate shells, commonly flattened and partially to entirely preserved, are a few millimetres thick but brittle and detached easily from their inner and outer mould. Their cream to white colour contrasts with the dark aspect of the sediment, and few thicker individuals are iridescent.

Molluscs shells from Warcq, clearly show a more energetic environment as they mostly appear as packed shell fragments showing a higher taxonomic diversity relative to the other site. Few complete individuals and separated valves can be observed among the debris with their associated mould in or around the remaining shell. Shells are cream to clear white, with some showing iridescence.

The remnants of carbonate shells were sampled as a whole using dental tools under a binocular-microscope. A carbonate vein and matrix from the carbonate nodule Pol 29 were also sampled to constrain the geochemistry of this potential diagenetic phase.

The microstructural preservation state and mineralogy of the analysed bivalve and ammonite shells were investigated using a Phenom Pure G2 scanning electron microscope (SEM) in backscatter mode and Raman spectroscopy using an XploRA Raman microscope in Laboratoire de Géologie de Lyon (LGL-TPE). SEM observations were performed on relatively large fragments of the most complete specimens. Raman spectra were acquired either directly on the fossil specimens partly enclosed in the sedimentary matrix, or on several grains of the sampled powders.

### 3.2 Geochemical analysis and data processing

The $\Delta_{47}$ and $\delta^{18}O$ values of 15 samples were measured (1 to 5 replicates each) using methods described by Daëron et al., (2016). Carbonate samples were converted to $CO_2$ by phosphoric acid reaction at 90 °C in a common, stirred acid bath for 15 minutes. Initial phosphoric acid concentration was 103 % (1.91 g/cm3) and each batch of acid was used for 7 days. After cryogenic removal of water, the evolved $CO_2$ was helium-flushed at 25 mL/min through a purification column packed with Porapak Q (50/80 mesh, 1 m length, 2.1 mm ID) and held at −20 °C, then quantitatively recollected by cryogenic trapping and transferred into an Isoprime 100 dual-inlet mass spectrometer equipped with six Faraday collectors (m/z 44–49). Each analysis took about 2.5 hours,

during which analyte gas and working reference gas were allowed to flow from matching, 10 mL reservoirs into the source through deactivated fused silica capillaries (65 cm length, 110 μm ID). Every 20 minutes, gas pressures were adjusted to achieve m/z = 44 current of 80 nA, with differences between analyte gas and working gas generally below 0.1 nA. Pressure-dependent background current corrections were measured 12 times for each analysis. All background measurements from a given session are then used to determine a mass-specific relationship linking background intensity ($Zm$), total m/z = 44 intensity ($I44$), and time ($t$): $Zm = a + bI44 + ct + dt2$. Background-corrected ion current ratios (δ45 to δ49) were converted to $\delta^{13}C$, $\delta^{18}O$, and "raw" $\Delta_{47}$ values as described by Daëron et al., (2016), using the IUPAC oxygen-17 correction parameters. The isotopic composition ($\delta^{13}C$, $\delta^{18}O$) of our working reference gas was computed based on the nominal isotopic composition of carbonate standard ETH-3 (Bernasconi et al., 2018) and an oxygen-18 acid fractionation factor of 1.00813 (Kim et al., 2007). Raw $\Delta_{47}$ values were then converted to the I-CDES $\Delta47$ reference frame by comparison with four "ETH" carbonate standards (Bernasconi et al., 2021) using a pooled regression approach (Daëron, 2021). Full analytical errors are derived from the external reproducibility of unknowns and standards (Nf = 89) and conservatively account for the uncertainties in raw $\Delta_{47}$ measurements as well as those associated with the conversion to the "absolute" $\Delta_{47}$ reference frame.

Complementary $\delta^{13}C$ and $\delta^{18}O$ analyses of the smallest Arctic shells were performed at LGLTPE, using a Multiprep$^{TM}$ automated sampler coupled to a dual-inlet GV Isoprime$^{TM}$ mass spectrometer. Samples were reacted with anhydrous phosphoric acid at 90°C. Duplicated samples were adjusted to the international references NIST NBS 18 and NBS 19 as well as in-house standard Carrara Marble. Since 2019 overall reproducibility of the in-house standard Carrara Marble are ±0.088‰ for $\delta^{18}O$ (2 SE, n = 649) and ±0.064‰ for $\delta^{13}C$ (2 SE, n = 649) with mean $\delta^{18}O$ and $\delta^{13}C$ values, respectively of - 1.041 ‰ and +2.025 ‰ (VPDB). All carbonate isotopic values ($\delta^{13}C$, $\delta^{18}O_c$) are reported in ‰ VPDB.

Clumped isotope temperatures were computed based on the I-CDES calibration of (Anderson et al., 2021). Temperature uncertainties correspond to the fully-propagated 95% confidence intervals from $\Delta_{47}$ measurements of each sample (Daëron, 2021), neglecting the much smaller uncertainties in the calibration. The $\delta^{18}O$ values from aragonite samples were adjusted considering the different phosphoric acid fractionation factors for calcite and aragonite (Kim et al., 2007). The $\delta^{18}O_w$ values relative to VSMOW was estimated using $\Delta_{47}$-derived temperatures and the equations of Grossman and Ku (1986) and Kim and O'Neil (1997) for mollusc shells and calcite vein respectively.

Paleolatitude of the studied sites was computed using the online paleolatitude calculator paleolatitude.org (van Hinsbergen et al., 2015) computed with the model of Torsvik et al. (2012).

## 4 Results

### 4.1 Polovinnaya River, Siberia

The SEM observations of shell fragments of *Dacryomya jacutica* revealed well-preserved sheet nacreous microstructures underlying a prismatic layer we interpret as the outer shell layer (Fig. 3). All Raman spectra gathered from *Dacryomya jacutica* shells confirm that the original aragonite mineralogy is preserved. $\Delta_{47}$ range from 0.6151±0.0108 to 0.6457±0.0182 ‰ I-CDES for Siberian aragonite bivalves, and a $\Delta_{47}$ of 0.5752±0.0134 for the fracture-infilling calcite vein. Reconstructed $\Delta_{47}$ temperatures, applying the equation of

Anderson et al., (2021), range from 8.8±5.2°C to 18.0±3.4°C for Siberian bivalves, and 31.5±4.8 °C for the calcite vein.

Mean $\delta^{18}O_c$ values are −2.73±0.71‰ (1SD, n=31, Max=0.36, Min=-5.08‰) for Siberian bivalves and −14.21±0.02 ‰ for the fracture-infilling calcite vein while the carbon isotope values ($\delta^{13}C$) range, from 3.47‰ to 5.09‰ in bivalve shells, and reach values down to −21.43‰ and −4.67‰ for carbonate nodule matrix and the embedded bivalve shells (sample POL-29) respectively. Using the $\Delta_{47}$-derived temperature to estimate the oxygen isotope fractionation factor results in $\delta^{18}O_w$ values ranging from −4.88±1.20‰ to −2.52±0.78‰ in Siberian bivalves. A much lower value of −10.6±0.9‰ is obtained for the fracture-infilling calcite vein from Polovinnaya River.

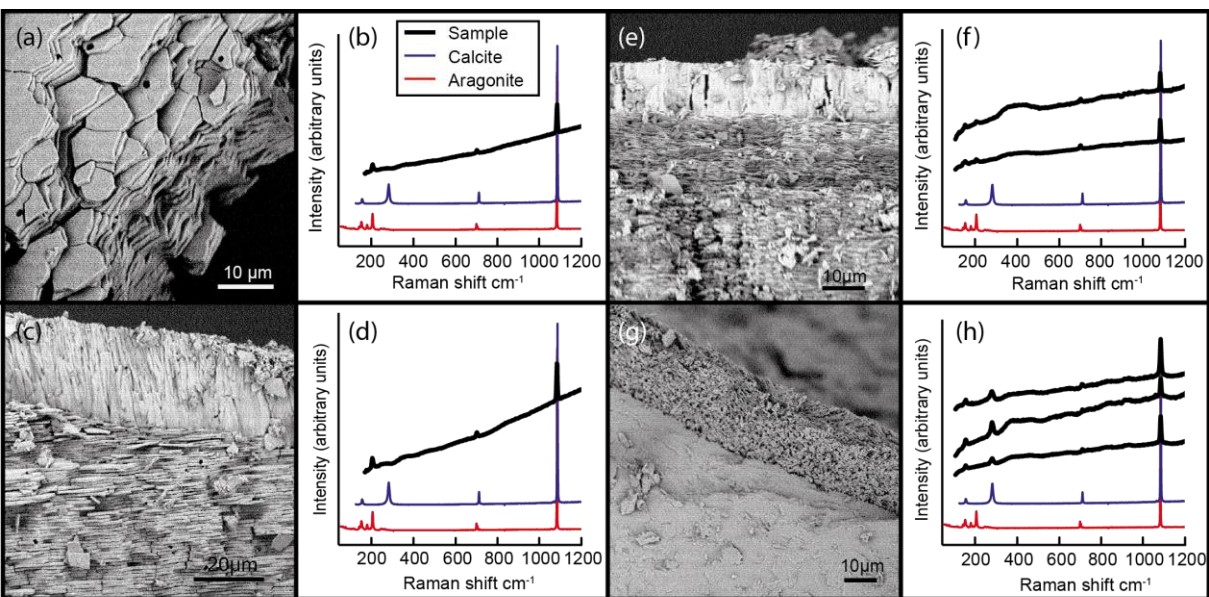

**Figure 3. SEM images and Raman spectra for a selection of samples.** a, c, e and g SEM images of samples POL-8, POL-12, ARD-05 and ARD-03 respectively. b, d, f and g Raman spectra of samples POL-8, POL-12, ARD-05 and ARD-03 compared to the reference spectra of calcite and aragonite.

**4.2 Warcq, France**

Mollusc shells from Warcq showing an aragonite mineralogy revealed microstructures similar to those observed in *Dacryomya jacutica* from Siberia, the main differences being that sheet nacreous structures of the studied ammonite shell (ARD-05) shows thinner tablets than those of bivalve shells (Fig. 3). Both SEM and Raman data indicate that the sample ARD-03 (bivalve fragment) is in calcite, showing a darker colour and no iridescence, with a much simpler and massive structure observed in SEM (Fig. 3).

$\Delta_{47}$ of measured bivalve and ammonite shells from Warcq range from 0.5851±0.0095 to 0.5955±0.0130 ‰ I-CDES. Reconstructed $\Delta_{47}$ temperatures range from 24.4±4.4°C to 28.0±3.3°C. Mean $\delta^{18}O_c$ values are −2.30±0.76‰ (1SD, n=6, Max=−0.83‰, Min=−2.79‰), and $\delta^{13}C$ range from 0.37‰ to 2.82‰. The calculated $\delta^{18}O_w$ values range from 0.6±0.7‰ to −1.4±1.0‰.

## 5 Discussion

### 5.1 Sample preservation

The SEM and Raman observations reveal that the analysed mollusc shells from both sites retain pristine aragonite mineralogy and microstructures with neither evidence of recrystallization nor mineralogical conversion (Fig. 3). Despite their aragonite mineralogy, the *Dacryomya* shells from sample Pol-29 record unusually low $\delta^{13}C$ that are ~8‰ lower than the other *Dacryomya* shells analysed from the same succession. The carbonate matrix of the nodule where these shells are embedded also records a very low $\delta^{13}C$ value (−21.43‰) but a $\delta^{18}O$ value within the range of the bivalve shells. We therefore attribute the extremely low $\delta^{13}C$ values of bivalves shells of this level to an early diagenetic phase resulting in the formation of carbonate nodules derived from respiratory $CO_2$ that locally altered the bivalve shells geochemistry.

Organic matter maturity, mineralogical and sedimentological data all imply exceptionally shallow burial depth (<1 km) for the samples investigated here. Maximum burial temperature ($T_{burial}$) remained well below the commonly assumed minimum temperature (80-120°C) of solid-state reordering of C-O bonds in calcite at geological timescales (Henkes et al., 2014; Stolper and Eiler, 2015; Hemingway and Henkes, 2021). Recent experiments show that aragonite is more susceptible to solid-state bond reordering (Chen et al., 2019), but there is to our knowledge no existing model constraining the temperatures at which this process would markedly overwrite the $\Delta_{47}$ value of this mineral at such time scales. The exceptional preservation of aragonite nacreous sheet microstructures in these samples implies minimal amounts of fluid circulation and recrystallization, if any. Exchange between fluid inclusions in mollusc shells and the surrounding carbonate minerals was recently suggested as an alternative process that may alter the clumped isotope signature of biogenic carbonates without substantially affecting the stable isotope signature of the shell nor its mineralogy (Nooitgedacht et al., 2021). In their heating experiments, these exchanges resulted in a significant decrease of the $\Delta_{47}$ value of the bivalve shells compared to the original shell, and a minor (~0.1‰) decrease in $\delta^{18}O$ of the heated shell. We cannot exclude that this process has altered the fossils studied here even at low temperature, nor do we have evidence that it occurred. The $\Delta_{47}$ temperature of 31.1±4.8°C for the fracture-infilling calcite vein in Arctic Russia is significantly higher than those inferred from bivalves and is consistent with a formation depth <1 km assuming a geothermal gradient of 25°C/km. The reconstructed $\delta^{18}O_w$ value of −10.7±0.9‰ for this calcite vein is also substantially lower than those inferred from associated bivalves, consistent with a late-phase meteoric source for the mineralizing fluid. The precise depth and date at which this vein formed however, remain uncertain.

Based on the geological setting of the samples and their preservation, we consider any substantial alteration of their original geochemical signature unlikely. In the remote scenario that the studied material has been slightly modified by solid-state bond reordering, one would expect the $\Delta_{47}$ of the samples to be lower than their original values (Henkes et al., 2014; Stolper and Eiler, 2015; Fernandez et al., 2021; Hemingway and Henkes, 2021). Therefore, both the temperature and $\delta^{18}O_w$ reported here would be overestimated, and should be taken as an upper limit of original environmental parameters.

**5.2 Evidence for extreme warmth and reduced salinity in the Arctic during the Toarcian Oceanic Anoxic Event**

Bivalve shells record a marked rise in $\delta^{13}C$ along the section up to ~5‰ that parallels that recorded by organic carbon $\delta^{13}C$ data (Fig. 4; Suan et al., 2011). These results strengthen the correlation of the corresponding part of the succession with the rising limb of the positive carbon isotope excursion commonly used to characterise the termination of T-OAE interval in coeval sites of Europe and North Africa (Jenkyns and Clayton, 1986; Suan et al., 2010; Krencker et al., 2014; Baghli et al., 2020; Ullmann et al., 2020). Bivalve shells $\delta^{18}O_c$, however, show no stratigraphic trend as opposed to brachiopod shell T-OAE records from western Tethys at mid-latitudes (Suan et al., 2010; Krencker et al., 2014; Ullmann et al., 2020; Baghli et al., 2020). Our $\Delta_{47}$ results yield polar temperatures ranging from 8.8±5.2 to 18.0±3.4 °C (Mean=14.7°C). As it occurs with most $\Delta_{47}$-derived temperature datasets, the relatively large uncertainties of the present estimates of Siberian SST hamper the identification of distinctive stratigraphic trends.

Bivalve shell growth can be highly variable during the animal life (Schöne, 2008), making any paleoenvironmental record derived from bivalve shell either incomplete (because of growth cessation) or at least biased towards the period of maximum growth rate. Shell growth rate can be controlled by both environmental parameters (Temperature, salinity, food availability ...), biological processes such as spawning, and changes during the ontogeny (Schöne, 2008). One major aspect of shell growth that may bias the geochemical signal data is seasonal shell growth-cessation. In modern high-latitude bivalves, seasonal shell growth-cessation generally occurs during the winter, triggered by low temperatures or low food availability (Peck et al., 2000; Vihtakari et al., 2016; Killam and Clapham, 2018). In the present-day *Nucula annulata*, an aragonite bivalve with similar ecology to the analysed *Dacryomya jacutica*, growth cessation occurs in winter and during spawning at peak local temperatures, its average $\delta^{18}O_c$ hence recording late spring to early fall SST (Craig, 1994). By contrast, growth band $\delta^{18}O_c$ offers evidence for summertime-only growth cessation in high-latitude Eocene bivalves from Antarctica, with inferred winter SST of 11.1±0.6 and summer SST of 17.6±1.3°C (Buick and Ivany, 2004; Douglas et al., 2014). A comparable seasonal $\delta^{18}O_c$ record could not be generated from our Russian Arctic material owing to the very small size of the available *Dacryomya* shells (1 to 2cm). In any case, the temperate data from NE France should be minimally affected by seasonal biases as shell precipitation occurs more continuously throughout the year in modern temperate molluscs (Killam and Clapham, 2018). Besides, both sites were deposited in near-shore environments at very shallow depths likely not exceeding a few tens of meters (Suan et al., 2011; Thuy et al., 2011). Although bivalves from both sections record temperatures near the sea bottom, that were likely slightly cooler than the sea surface, the difference should not have exceeded a few degrees, owing to their shallow living depth we expect the studied bivalves to have lived within the thermocline. We therefore conservatively interpret the reconstructed temperatures as reflecting polar warm-season SST (summer; $SST_{PWS}$) in Arctic Russia and low latitude annual SST in NE France. These $SST_{PWS}$ for the T-OAE are still 10-20°C higher than present-day $SST_{PWS}$ (Fig. 5).

The reconstructed polar $\delta^{18}O_w$ values ranging from –4.9 ±1.2 to –2.5 ±0.8‰ VSMOW during the T-OAE are significantly lower than the value of –1‰ VSMOW expected for an ice-free world mean open ocean (Shackleton and Kennett, 1975). These results imply a substantial freshwater contribution to the studied basin during the T-OAE, probably resulting from coastal runoff at this relatively proximal site (Suan et al., 2011). High temperatures and reduced salinity are in broad agreement with paleontological evidence for warm and humid

temperate conditions during the T-OAE interval in Arctic Siberia (Rogov et al., 2019). Brackish conditions are also consistent with the fossil assemblages of the succession that includes abundant terrestrial organic matter and wood debris, marine to brackish elements such as abundant dinoflagellate cysts, benthic foraminifera (preserved as organic linings and agglutinate forms) and typically marine elements that are represented by a few belemnite rostra and unidentifiable ammonite internal moulds (Suan et al., 2011). Interestingly, protobranch bivalves, to which *Dacryomya* belongs, are not well adapted to salinities lower than 20‰ (Zardus, 2002). Assuming a similar lower salinity limit for Polovinnaya River bivalves, a global mean ocean with a salinity of 34.5‰ and a $\delta^{18}O_w$ of –1‰ VSMOW, mass balance considerations (see Supplementary Data) imply an upper limit of ∼−8‰ VSMOW for local $\delta^{18}O$ of precipitations and runoff ($\delta^{18}O_p$). This value is high relative to modern Arctic $\delta^{18}O_p$ but in agreement with the prediction that higher polar temperatures should have produced higher $\delta^{18}O_p$ than those prevailing today (Rozanski et al., 1992). Similarly, terrestrial plants *n*-alkanes hydrogen isotopes and paleosol siderite $\Delta_{47}$ data indicate slightly lower Arctic $\delta^{18}O_p$ of –10 to –15‰ VSMOW during the Early Eocene (Pagani et al., 2006; van Dijk et al., 2020), another well-established warm period with evidence of polar warmth (Markwick, 1998; Sluijs et al., 2006, 2020; Douglas et al., 2014; Suan et al., 2017; van Dijk et al., 2020). Assuming a similar range of $\delta^{18}O_p$ values in the Early Jurassic Arctic, and the assumption listed above for the Early Jurassic oceans, mass balance calculations indicate mean salinity of 23.9±2.9‰ (1σ, n=8) and 27.7±1.8‰ (1σ, n=8) with $\delta^{18}O_p$ of –10 and –15‰ VSMOW, respectively (see Supplementary Data), again consistent with paleontological evidence suggesting brackish waters at Polovinnaya River during the Toarcian. Such values are also comparable to the salinity of 28‰ estimated using a fully coupled ocean–atmosphere model for the Toarcian (Dera and Donnadieu, 2012), although the Arctic temperature obtained by the same model are in strong disagreement with our data (Fig. 5). Such observations should be replicated around the Arctic realm to test whether the brackish environment evidenced here was of local or more regional nature.

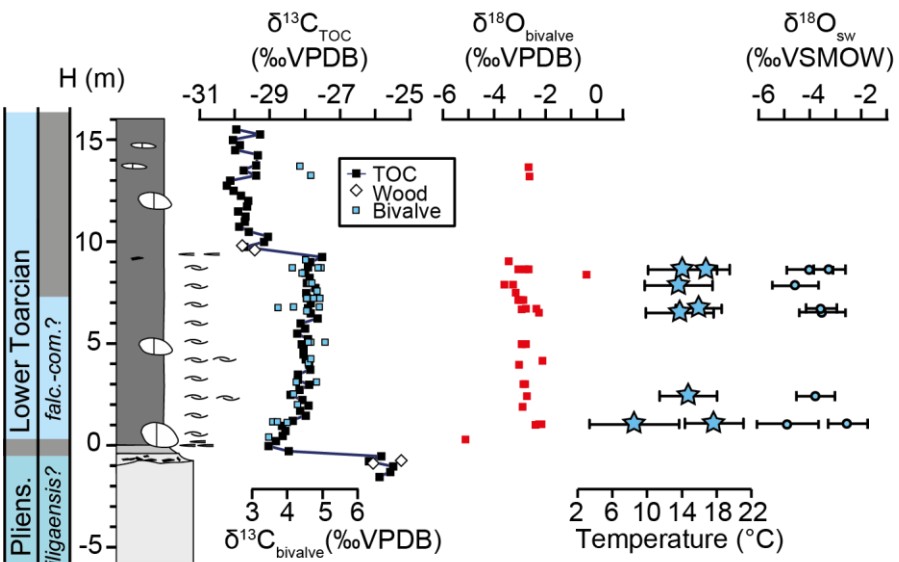

**Figure 4. Geochemical record of the T-OAE at Polovinnaya River, Arctic Siberia.** Stratigraphy and biostratigraphic zones are for the Arctic realm (falc.=Falciferum zone, com.=Commune zone). The organic carbon isotope ($\delta^{13}C_{TOC}$) data (black squares) wood debris $\delta^{13}C$ data (white diamonds) are from Suan et al. (2011); the bivalve shell $\delta^{13}C$ data (blue squares), bivalve shell $\delta^{18}O$ (red squares), $\Delta_{47}$ temperatures (blue stars), and $\delta^{18}O_w$ estimates (blue circles) inferred from bivalve shells $\Delta_{47}$ values are from this study. The analysed bivalve samples all belong to the species *Dacryomya jacutica*. $\Delta_{47}$-derived temperatures were computed using

the equation of Anderson et al. (2021). $\delta^{18}O_w$ was calculated using the oxygen isotope fractionation equation of Grossman and Ku (1986).

**5.3 Early Jurassic latitudinal temperature and $\delta^{18}O_w$ gradients**

The mid-paleolatitude SST reconstructed by our new clumped isotope data (~25°C) are in good agreement with recent Sinemurian-Pliensbachian and Toarcian $TEX_{86}^H$ data pointing to summer SST~20-30°C at slightly lower paleolatitudes (Robinson et al., 2017; Ruebsam et al., 2020). It should be noted that the mid-latitude samples presented here are from the Davoei Zone and predate the T-OAE interval recorded by the Siberian data by ~6 million years. Nevertheless, climate proxies from the Davoei zone indicate the corresponding time interval, although likely slightly cooler than the T-OAE, correspond to one of the warmest period of the Early Jurassic (Dera et al., 2011; Bougeault et al., 2017). The new clumped isotope data from the two sites, even if they are not strictly contemporaneous, can therefore be reasonably used to tentatively estimate latitudinal gradient during the warmest episodes of Early Jurassic. The $\Delta_{47}$ data presented herein suggest a decrease in mean SST of 0.26±0.05°C per ° of latitude between mid and high latitudes, i.e., a reduction of the latitudinal SST gradient of 32±10% relative to present, consistent with the most conservative Early Eocene estimates (Evans et al., 2018). Comparing our Siberian T-OAE $\Delta_{47}$ temperatures with contemporaneous $TEX_{86}^H$ temperatures estimated for low latitudes (Ruebsam et al., 2020) results in an even shallower gradient of 0.17±0.05°C per ° of latitude between low and high latitudes.

Considering the scarcity of other Early Jurassic temperature proxy-data, model-based SST and $\delta^{18}O_w$ estimates, we extend the comparison to SST and $\delta^{18}O_w$ estimates based on various proxy-data and published Earth system simulations for other Jurassic to Eocene intervals (Fig. 5; Supplement). First, the new $\Delta_{47}$ temperatures can be compared with other well established warm intervals, such as the Cenomanian-Turonian and the Eocene. The compilation shows that $\Delta_{47}$ SST from NE France agree with most previous mid-latitude $TEX_{86}^H$ and $\Delta_{47}$ SST for the Eocene interval (20-30°C), with values > 5°C higher than present-day SST. Mid-latitude Cenomanian-Turonian $TEX_{86}^H$ SST are significantly higher (>30°C) and form the highest mid-latitude temperature of the compilation. High latitude data are much more scarce. Still, Russian Arctic Toarcian SST are very close to Early Eocene polar SST derived from Arctic (Sluijs et al., 2006, 2020; Suan et al., 2010) and Antarctic (Douglas et al., 2014) sites with polar SST >15°C warmer than present during these two distinct greenhouse period (Fig. 5). Our $\Delta_{47}$ SST for the Lower-Jurassic can be compared to published results from Earth System models that simulate these intervals of global warmth (see previous paragraph) to discuss model-data discrepancies, especially apparent at high latitudes. Proxy-data indicate an atmospheric $pCO_2$ of 1000±500 ppmv during the Early Jurassic, with maximum values of 1750±500 ppmv, i.e., 6x pre-industrial levels (PIL), during the T-OAE (McElwain et al., 2005; Li et al., 2020). Earth system models run at 6x PIL for Early Jurassic (Dera and Donnadieu, 2012) or Cretaceous-Eocene paleogeography almost invariably produce lower SST than those inferred from our $\Delta_{47}$ data, with a maximum model-data discrepancy of >15°C at high latitudes (Fig. 5). To achieve such polar warmth, the Eocene CCSM3 simulations require 16x PIL, more than twice that indicated by Lower Jurassic and Eocene proxy data (Huber and Caballero, 2011). Reconstructed SST of 14.4±2.8°C near the North Pole during the T-OAE, however, correspond to the maximum monthly temperatures simulated by the Turonian IPSL-CM5A2 model near the North Pole at 4x PIL (Laugié et al., 2020). The hypothesis of shell growth restricted to warmest month in the analysed Toarcian Arctic bivalves, however, remains questionable

given the evidence for summertime-only growth cessation in Eocene bivalves from Antarctica (Buick and Ivany, 2004). Finally, Arctic SST as high as 15-20°C are successfully achieved in the Eocene CESM1.2 CAM5 at 6 to 9x PIL (Zhu et al., 2020), in which climate sensitivity increases with rising $CO_2$ due to low-altitude cloud albedo feedbacks and improved radiative parameterization. As this model produces an increase in climate sensitivity with $CO_2$ in both Eocene and modern conditions, our results thus support the growing body of evidence that the amplitude of the future anthropogenic warming may be underestimated by conventional state-of-the-art models.

The reconstruction of $\delta^{18}O_w$ values using proxy data provides a complementary aspect to assess model capabilities, as this indicator is sensitive to both climate parameters (moisture, humidity and temperatures) and paleogeography. Our mid-latitude $\delta^{18}O_w$ are broadly similar to those reconstructed using marine turtle bones $\delta^{18}O_{PO4}$ and $\Delta_{47}$ data from Jurassic to Eocene bivalves, ammonites and foraminifera (Fig. 5, Billon-Bruyat et al., 2005; Coulson et al., 2011; van Baal et al., 2013; Evans et al., 2018; Wierzbowski et al., 2018; Vickers et al., 2021; de Winter et al., 2021). The reconstructed $\delta^{18}O_w$ values are, to some extent, also broadly comparable with those inferred from belemnite calcite $\Delta_{47}$ data (Wierzbowski et al., 2018; Vickers et al., 2019, 2020, 2021; Price et al., 2020), although such data should be interpreted with caution owing to the likely unique oxygen isotope fractionation of belemnite calcite (Price et al., 2020; Vickers et al., 2021). In line with evidence for substantial [18]O-depletion of Toarcian Arctic waters (relative to VSMOW) suggested by our data, previous studies suggested low $\delta^{18}O_w$ values in interior seas border by large continental areas, such in the Western Interior Seaway during the Campanian-Maastrichtian (Coulson et al., 2011; Petersen et al., 2016b; Meyer et al., 2018) and Middle Russian sea during Middle to Late Jurassic interval (Wierzbowski et al., 2018). Interestingly, the $\Delta_{47}$ temperatures reported in these basins are also differ markedly from those reported in more open-oceans sites of similar age and paleo-latitude, suggesting the possible influence of colder Arctic water masses through southward ocean currents. Indeed the Middle to Late Jurassic $\Delta_{47}$ temperatures reported in the Middle Russian sea (Wierzbowski et al., 2018) are ~10°C lower than coeval data from the Hebrides basin, Scotland (Vickers et al., 2020) and from the high mid-latitudes Falkland Plateau in the southern hemisphere (Vickers et al., 2019). Such anomalies in $\delta^{18}O_w$ values and temperatures demonstrate the importance of regional pattern such as river runoff and basin connections on the environmental parameters of restricted basin (Petersen et al., 2016b). More generally, local anomalies in $\delta^{18}O_w$ values evidenced by the new and earlier clumped isotope data highlight ability of this proxy of deciphering the influence of the temperature and $\delta^{18}O_w$ values in otherwise similar $\delta^{18}O_c$ dataset as well as the.

We are aware of only three Earth-system $\delta^{18}O_w$ simulations for the broad time interval considered here (Zhou et al., 2008; Tindall et al., 2010; Zhu et al., 2020), hence limiting model-data comparisons. The higher freshwater contribution near high-latitude landmasses of the Northern Hemisphere in all these models produced lower $\delta^{18}O_w$ that are broadly consistent with previous and our proxy data (Fig. 5). This good agreement, however, might be partly fortuitous, as proxy data suggest SST much higher than those produced by these models (Fig. 5). As mentioned above (section 5.2), such higher-than-predicted polar warmth would have substantially increased high-latitude $\delta^{18}O_p$, so that higher runoff would be required to reproduce the magnitude of the poleward drop in $\delta^{18}O_w$ indicated by proxy data. This highlights the usefulness, in future models of past greenhouse climates, to systematically provide $\delta^{18}O_w$ predictions so that $\delta^{18}O_w$ estimates derived from $\Delta 47$ data may serve as a constraint on Earth system models.

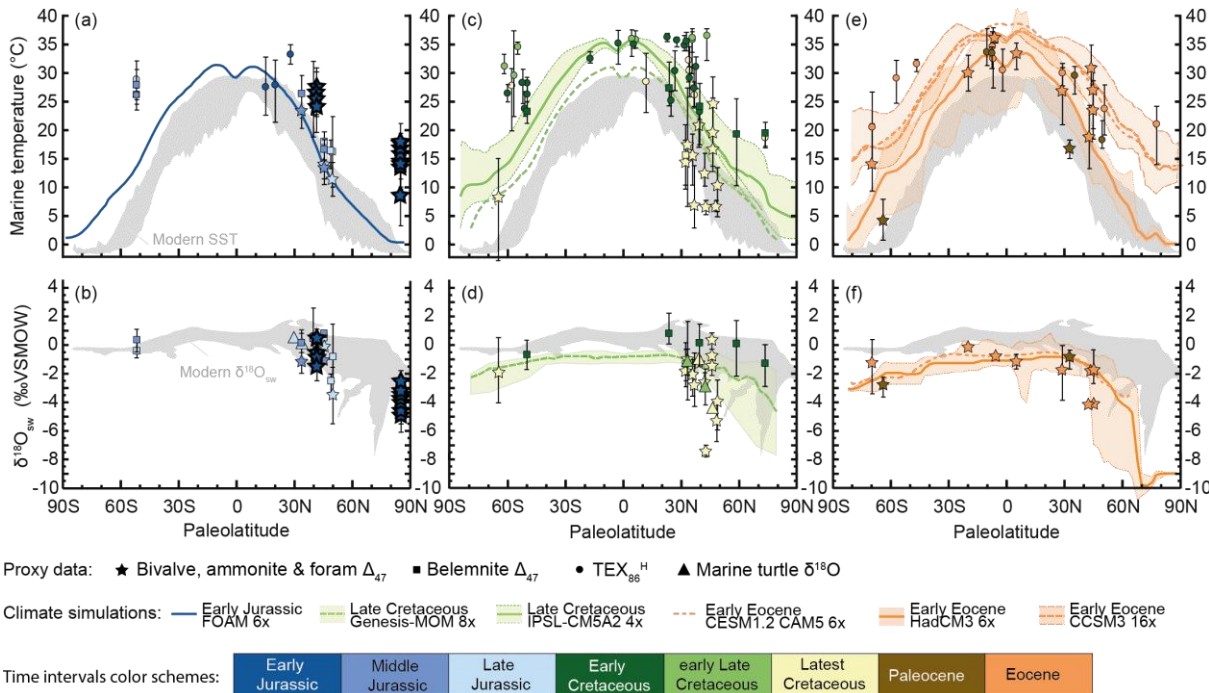

**Figure 5.** Comparison of the new (bold outline) reconstructed Early Jurassic SST and $\delta^{18}O_{sw}$ with published Jurassic-Eocene proxy-based reconstructions (thin outline) and Earth system simulations. Proxy-model comparison of SST and $\delta^{18}O_{sw}$ are shown for the Jurassic (a, b), the Cretaceous (c, d) and the Early Paleogene (e, f). Proxy data are divided between 8 time slices based on their definition in the International Chronostratigraphic chart v2020/03 (Cohen et al., 2013; updated); early Late Cretaceous=Cenomanian to Santonian; Latest Cretaceous=Campanian-Maastrichtian. Marker colour shows sample age accordingly (see key). New data are displayed as the reported value and its associated uncertainties for each samples. Datasets from the literature, from the same proxy and location, were regrouped within each time slice. The marker displays the mean of available data and the error bar the extent of the data (minimal to maximal value). Both $\Delta_{47}$ and $TEX_{86}^H$ temperatures are published temperatures. $\delta^{18}O_w$ were recomputed using published $\Delta_{47}$ temperatures and $\delta^{18}O_c$ with the following fractionation equations: belemnite calcite: Coplen, (2007), aragonite: Grossman and Ku(1986), bivalve calcite: Epstein et al. (1953), foraminifera: Erez and Luz (1983), turtle bones: Barrick et al., (1999) updated (Pouech et al., 2014). Results of Earth system simulations from the literature are shown as annual averages (bold lines) and summer and winter seasonal averages (colour shading). Modern range of SST and $\delta^{18}O_{sw}$ are also shown (grey shading) for comparison. $\Delta_{47}$ data are from: Keating-Bitonti et al., 2011; Douglas et al., 2014; Petersen et al., 2016b, a; Evans et al., 2018; Wierzbowski et al., 2018; Meyer et al., 2018; Vickers et al., 2019, 2020, 2021; Price et al., 2020; Brigaud et al., 2020; Fernandez et al., 2021; de Winter et al., 2021. $TEX_{86}^H$ paleothermometry data are from: Jenkyns et al., 2012; Lunt et al., 2012; Douglas et al., 2014; Frieling et al., 2014; O'Brien et al., 2017; Robinson et al., 2017; Cramwinckel et al., 2018; O'Connor et al., 2019; Ruebsam et al., 2020; Cavalheiro et al., 2021 . Marine turtle phosphate $\delta^{18}O$ data are from : Billon-Bruyat et al. (2005), Coulson et al. (2011), van Baal et al. (2013). The FOAM 6x simulation is from Dera and Donnadieu (2012), Genesis-MOM 8x simulation is from Zhou et al. (2008), IPSL-CM5A2 4x is from Laugié et al. (2020), CESMA.2 CAM5 6x is from Zhu et al. (2020), HadCM3 6x is from Tindall et al. (2010), CCSM3 16x is from Huber and Caballero (2011), with Nx indicating $CO_2$ levels used in the simulations as a multiple of preindustrial

level (i.e. 280ppm). More detailed information on the building of this figure and the data compilation are available as Supplementary material.

## 6. Conclusion

The clumped isotope compositions of pristine, minimally buried, marine mollusc shells yield SST >25°C at mid-latitudes during the early Pliensbachian and SST >10°C at polar paleolatitudes during the T-OAE. The reconstructed $\delta^{18}O_w$ values point to higher freshwater contribution toward Arctic regions, illustrating the dangers of assuming a fixed global $\delta^{18}O_w$ value for $\delta^{18}O$-derived temperature reconstructions. Although further work should clarify the influence of seasonal changes in the recorded SST values at polar sites, these results strengthen a growing body of evidence for higher climate sensitivity under high atmospheric $CO_2$ conditions and suggest that this higher sensitivity is a general feature of greenhouse climates since at least 180 Ma.

**Data availability**

Detailed data supporting this study are available in the supplementary material. Raw data are available on request to the author.

**Author contribution**

TL and GS designed the study and led the writing in close cooperation with CL, MR and MD. MR and GS participated to the field work and collected the samples. TL prepared and sampled the shell material for geochemistry and performed the SEM observations. MR, JS and OL identified the fossils. MD and TL performed the clumped isotope analyses and data processing. AV-L and TL performed the stable isotopes analyses and data processing. BR, GM and TL gathered and interpreted the Raman spectra. TL and GS compiled the paleotemperature proxy database. All authors were involved in the interpretation of the results.

**Competing interest**

The authors declare that they have no conflict of interest.

**Acknowledgments**

This research was funded by the ANR OXYMORE (ANR-18-CE31-0020), joint CNRS/RFBR International Emerging Action grant 205700 and RFBR grant 21-55-15015. JS was supported by grant APVV 17-0555. We thank Ghislaine Broillet for her help with SEM analyses, and Ophélie Lodyga for her help with Raman analyses. We thank two anonymous reviewers for their constructive suggestions and comments that substantially improved the manuscript.

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
