# Peer review of "Clumped isotope evidence for Early Jurassic extreme polar warmth and high climate sensitivity"

_Climate of the Past, 2021_

## Author Comment (AC1)

**Response to Referee #1**

First we would like to thank the reviewer for the time invested to review our paper and constructive comments and suggestions. Comments that will lead to substantial modification of the manuscript are discussed below. As for the more specific minor corrections, they will be addressed in the revised version of the paper.

**General comments**

*This is an interesting paper that adds to the debate on the problematic nature of apparent polar warmth during at least some intervals during the Mesozoic greenhouse. In this context, the data from Siberia are particularly valuable, particularly as a number of workers are insisting on the presence of substantial Jurassic and Cretaceous ice to explain sea-level changes and cold-climate phenomena such as glendonites in high-latitude sites. The fact remains, of course, that the present work offers only a snapshot of geological time, in the Toarcian case during a well-established hyperthermal, and extreme extrapolation to much of the Mesozoic would probably be unwise.*

We agree with this statement. Obviously the data from lower Toarcian, specifically the TOAE cannot describe what occurred during colder Mesozoic intervals such as the late Pliensbachian, the Bajocian-Bathonian or the early Aptian. The main comparison was first made with the Early Eocene or the Cenomanian-Turonian transition, other well-established warm intervals, hence the comparison with climate models performed for such periods. Yet we also wanted to compare-it to the rest of the Mesozoïc to insert the early Toarcian within this climate history. The above comments suggest we must clarify the discussion in this sense. We propose to be more specific on the climate mode estimated for each interval used in this comparison, in order to better point the issue of polar warmth recorded during hyperthermal events, and the struggle of climate models to achieve such polar warmth.

*Given the importance of the Arctic data, I think it would be preferable in parts of the text (e.g. Results and Discussion) to separate out the Pliensbachian and Toarcian data sets in separate subsections rather than running them together, which can become confusing to the reader.*

We agree with this suggestion will address this issue for the revised manuscript.

*In terms of fidelity of the paleotemperature records, much depends on the preservational state of the aragonitic fossils, and the authors have made some obvious moves to determine the integrity of their material. I must say, however, that, from the photographs, the Arctic specimens have a white 'powdery' look to them, which is typical for partly degraded aragonite.*

Indeed, the fidelity of the record depends on the preservation of the fossil material. The superficial (optical) aspect of the shell can be dramatically altered by the mechanical crushing of the shell, which is common in such fine-grained sediments and clearly visible in SEM images for some samples. Besides, we did not find any evidence for mineralogical conversion in either the Raman spectra or the SEM observations.

*As another test of alteration, strontium-isotope data would be useful, since the Toarcian global curve has particularly low values around the OAE interval and the presence of more*

*radiogenic $_{87}Sr/_{86}Sr$ ratios would be a fingerprint for alteration. Ideally, of course, there would be some accompanying TEX86 data, which should be obtainable given the relative lack of maturity of the sediments and at least a modest amount of organic material in the sediment.*

Performing strontium isotope analysis on these samples would indeed provide interesting insights into their preservation state, provided the primary values have not been influenced by the presumably high freshwater input of radiogenic strontium in the first place. This is something we plan to do. However, for logistic reasons, we cannot perform such analysis in a near future (not before early 2022), so this is for-now out of the scope of our study. The same is true for TEX86; however, there is a slight chance that GDGTs are preserved here, and again, this looks instead as a task for further, dedicated study.

**Specific comments**

*Line 244: can 'only a few degrees' be more specific? Estimates of the temperature drop across the thermocline from some localities during the Jurassic and Cretaceous, based on belemnite delta-18O values and TEX 86, come in at about 14°C (Mutterlose et al., Earth and Planetary Science Letters, 298, 286-298 and Jenkyns et al., Climate of the Past, 8, 215-226). So, presumably the bivalves were living in the mixed layer? As noted above, it would be useful to have some TEX86 values for the accompanying sediments.*

From data derived in other Siberian sections, Dacryomya-Tancredia–dominated assemblages were common in relatively deep but near-shore environments (Shurygin, 2005). (Zakharov and Shurygin, 1978) referred Dacryomya to as eurybathic infaunal deposit feeder tolerant to low oxygen contents, which prefer environments with slow hydrodynamics. Position of natural habitat of these bivalves in relation to thermocline remains unclear. Dacryomya genus is one of the most common bivalve associated with the Toarcian OAE in fully marine facies around the World.

*Figure 4, text figure explanation. Please explain what the different symbols mean and the shorthand for the zones. Should falciferum not now be serpentinum?*

We will address the figure explanation, and detail the ammonite zone name. As for the question regarding ammonite zone, here the Siberian zonal succession is used, and it differs a little from the European zonation.

*Fig 5, text-figure explanation is not comprehensive enough, making this diagram difficult to decipher. Make clear what grey bands signify. References should be given here, not in Supplementary data.*

The caption will be clarified and references added in as suggested.

**References:**

Shurygin, B.N., 2005. Lower and Middle Jurassic Biogeography, Facies, and Stratigraphy in Siberia Based on Bivalve Mollusks. Geo, Novosibirsk.

Zakharov, V.A., Shurygin, B.N., 1978. Biogeography, facies and stratigraphy of the Middle Jurassic of Soviet Arctic (by bivalve molluscs). Transactions of the Institute of Geology and Geophysics, Siberian Branch of the Academy of Science of USSR 352, 1–206.

---

## Author Response (AR1)

**Author's response file.**

All line calls in this file refer to the Author's track change file.
It is presented as follow:
- Referee comments
- Author's response
- Changes to the manuscript

**Anonymous Referee #1**

*General comments*

This is an interesting paper that adds to the debate on the problematic nature of apparent polar warmth during at least some intervals during the Mesozoic greenhouse. In this context, the data from Siberia are particularly valuable, particularly as a number of workers are insisting on the presence of substantial Jurassic and Cretaceous ice to explain sea-level changes and cold-climate phenomena such as glendonites in high-latitude sites.

The fact remains, of course, that the present work offers only a snapshot of geological time, in the Toarcian case during a well-established hyperthermal, and extreme extrapolation to much of the Mesozoic would probably be unwise.

We agree with this statement. Obviously the data from lower Toarcian strata, specifically the TOAE, do not preclude much cooler polar conditions during other time intervals. The main comparison was first made with the Early Eocene or the Cenomanian-Turonian transition, other well-established warm intervals that have been targeted by climate model simulations. Yet we also wanted to compare our new results with those obtained for the rest of the Mesozoic to replace the new early Toarcian results within this climate history. The above comments suggest we must clarify the discussion in this sense. We propose to be more specific on the climate mode estimated for each interval used in this comparison, in order to better point the issue of polar warmth recorded during hyperthermal events, and the struggle of climate models to achieve such polar warmth.

Section 5.3 was substantially rearranged to clarify the comparisons between our new data, published proxy-data and climate models, by first comparing temperatures then $\delta^{18}O_w$.
For the discussion of temperatures, we first discuss latitudinal temperature gradients based on our data and other Toarcian proxy data, and then compare the obtained gradients with those reported for the Early Eocene, another well-established greenhouse period for polar temperatures have been reconstructed. We then compare our new data with earlier temperature proxy data from greenhouse intervals (Cenomanian-Turonian, Eocene) and to model simulations of these greenhouse periods.
The end of the discussion now begins with a general comparison of mid-latitude $\delta^{18}O_w$ data, with a special focus on regions where low $\delta^{18}O_w$ values comparable to our Toarcian Arctic $\delta^{18}O_w$ have been reported. The section ends with a proxy-model comparison of $\delta^{18}O_w$.

Given the importance of the Arctic data, I think it would be preferable in parts of the text (e.g. Results and Discussion) to separate out the Pliensbachian and Toarcian data sets in separate subsections rather than running them together, which can become confusing to the reader.

We agree with this suggestion will address this issue for the revised manuscript.

Both studied sites have now their own subsection in the result section (Section 4), as it was the case for the Geological setting (Section 2). Section 5.2 only deals with data from Siberia. The new data from the two time-intervals are then compared more critically than in the former version in section 5.3. The potential limitations of such comparisons are now stated explicitly

In terms of fidelity of the paleotemperature records, much depends on the preservational state of the aragonite fossils, and the authors have made some obvious moves to determine the integrity of their material. I must say, however, that, from the photographs, the Arctic specimens have a white 'powdery' look to them, which is typical for partly degraded aragonite.

Indeed, the fidelity of the record depends on the preservation of the fossil material. The superficial (optical) aspect of the shell can be dramatically altered by the mechanical crushing of the shell, which is common in such fine-grained sediments and clearly visible in SEM images for some samples. Besides, we did not find any evidence for mineralogical conversion in either the Raman spectra or the SEM observations.

**Line 100-105:** We have added details in the text figure explanation of Figure 2, by mentioning the specimens showing inner moulds after sampling rather than shells.
**Line 243-244:** We now state that the Raman spectra and SEM images show no evidence for neither recrystallization nor mineralogical conversion.

As another test of alteration, strontium-isotope data would be useful, since the Toarcian global curve has particularly low values around the OAE interval and the presence of more radiogenic $^{87}Sr/^{86}Sr$ ratios would be a fingerprint for alteration. Ideally, of course, there would be some accompanying $TEX_{86}$ data, which should be obtainable given the relative lack of maturity of the sediments and at least a modest amount of organic material in the sediment.

Performing strontium isotope analysis on these samples would indeed provide interesting insights into their preservation state, provided the primary values have not been influenced by the presumably high freshwater input of radiogenic strontium in the first place. This is something we plan to do. However, for logistic reasons, we cannot perform such analysis in a near future (not before early 2022), so this is for-now out of the scope of our study. The same is true for $TEX_{86}$; however, there is a slight chance that GDGTs are preserved here, and again, this looks instead as a task for further, dedicated study.

No changes needed as such data are still not available.

*Specific comments*

Line 24: change to 'These data highlight the risk . . .' Corrected

Line 60: should be 'as well as isolated' Corrected

Line 62: has undergone Corrected

Line 115: commonly flattened Corrected

Line 139: described by Daëron et al. (2016). Corrected

Line 141: $CO_2$ Corrected

Line 153: need subscripts Corrected

Line 165: values, respectively (insert comma) Corrected

Line 199: well-preserved Corrected

Line 285: say where in the mid-latitudes. We have added "western Tethys" to precise where.

Line 295: high-latitude Corrected

Line 301: hence recording . . . Corrected

Line 308: near-shore Corrected

Line 311: should not have exceeded Corrected

Lines 315–341: relative to SMOW standard. We have added "VSMOW" after the numerical values

Line 311: can 'only a few degrees' be more specific? Estimates of the temperature drop across the thermocline from some localities during the Jurassic and Cretaceous, based on belemnite delta-18O values and TEX 86, come in at about 14°C (Mutterlose et al., *Earth and Planetary Science Letters*, *298*, 286-298 and Jenkyns et al., C*limate of the Past*, *8*, 215-226). So, presumably the bivalves were living in the mixed layer? As noted above, it would be useful to have some TEX86 values for the accompanying sediments.

From data derived in other Siberian sections, Dacryomya-Tancredia–dominated assemblages were common in relatively deep but near-shore environments (Shurygin, 2005). (Zakharov and Shurygin, 1978) referred Dacryomya to as eurybathic infaunal deposit feeder tolerant to low oxygen contents, which prefer environments with slow hydrodynamics. Position of natural habitat of these bivalves in relation to thermocline remains unclear. Dacryomya genus is one of

the most common bivalve associated with the Toarcian OAE in fully marine facies around the World.

We now state that the studied bivalves likely lived within the thermocline.

Figure 4, text figure explanation. Please explain what the different symbols mean and the shorthand for the zones. Should *falciferum* not now be *serpentinum*?

We will address the figure explanation, and detail the ammonite zone name. As for the question regarding ammonite zone, here the Siberian zonal succession is used, and it differs a little from the European zonation.

We have added all abbreviations for the biostratographic zone and the explanation of the different symbols.

Line 362: mid-paleolatitude Corrected

Line 419-423: references required References added

Line 652: reference is incomplete Corrected

Fig 5, text-figure explanation is not comprehensive enough, making this diagram difficult to decipher. Make clear what grey bands signify. References should be given here, not in Supplementary data.

The caption will be clarified and references added in as suggested.

We have added the meaning of the grey band, and the list of references used in the compilation of geochemical proxy data and those associated with each displayed climate simulation. The green colour for Early Cretaceous data has been darkened to better differentiate Early Cretaceous data and early Late Cretaceous data.

**Anonymous Referee #2**

*General comments*

This manuscript describes a compelling dataset of carbon, oxygen, and clumped isotope ratios from late Jurassic fossil shells, one at northern polar latitudes in the Toarcian and another from mid-latitudes in the Pliensbachian. Their sub-period designation is about the only commonality between the samples and the data, yet the authors discuss them in nearly the same breath.

Line 364-371– We now explicitly state that we do not consider the two sites as contemporaneous, and the possible limitations of these comparisons. We have also added a sentence justifying the comparison of samples recording some of the warmest intervals of the Early Jurassic.

For this and other reasons articulated below I am recommending that this manuscript could dramatically improve following careful, major revisions. The major strengths of the paper are in the quality of the stable isotope data, which by all accounts is very high and uses the most up-to-date clumped isotope methods, correction schemes, and temperature calibration curves. However, I am routinely irked by the habitual reporting of clumped isotope temperatures to the tenth decimal place, which hardly seems to matter when the reported error is several degrees.

"Beyond this, I fear the manuscript falls into a common trope of being too quick to overlook the possibility of clumped isotope resetting during burial […]"

The possibility of clumped isotope resetting in our data was clearly stated in our MS, but we recognize that the sections dedicated to this aspect could be clarified and expanded in the revised version. The various available elements constraining sample burial and our current understanding of clumped isotope resetting all very limited clumped isotope resetting. The Rock Eval results from a previous study (L 60-61) indicates the organic matter is immature, thus constraining the upper limit of burial to the oil window. We agree that this information is not sufficient to prove that there was no clumped isotope resetting as we measured samples (unpublished data) from slightly more mature Arctic sites that indisputably shows resetting while below the oil window. This is why we also estimated the local burial based on available sedimentology data (Line 61 to 65) and present the local geothermal gradient to constrain the heat the samples could have undergone. Surely the geothermal gradient evolved during the thermal history of the samples but we make the approximation that it remained relatively low, as the site lies on the Siberian craton. For these reasons we estimate that the samples are very unlikely to have been substantially reset (again based on our current knowledge of clumped isotope resetting). Yet latter in the discussion, we cite the recent work of Nooitgedacht et al (2021) who propose that internal water can facilitate clumped isotopes reordering and explicitly declare that "We cannot exclude that this process altered the fossils studied here…" (Line 203).

For each site, we now estimate maximum burial temperature (Section 2). We now also state that aragonite is more prone to solid-state bond reordering than calcite (Section 5.1). We provide more details in the associated specific comments below.

" […] and too ready to extrapolate results across paleolatitudes and Phanerozoic timescales with grand paleoclimate ambition."

This remark, together with comments from the other anonymous referee, indicate that the comparison between the data presented in this manuscript and those from the literature may have been confusing to the readers. We will therefore substantially rework this section by focusing on the comparison of Toarcian data with other very warm periods (Cenomanian-Turonian, Early Eocene …) and made clearer that existing data indicate that the whole Mesozoic was not a uniform greenhouse period.

As suggested here, and as mentioned above for the comments of the other anonymous referee, Section 5.3 was substantially reorganized to clarify the comparisons between our new data, published proxy-data and climate models, by first comparing temperatures then $\delta^{18}O_w$. For the discussion of temperatures, we first discuss latitudinal temperature gradients based on our data and other Toarcian proxy data, and then compare the obtained gradients with those reported for the Early Eocene, another well-established greenhouse period for which polar temperatures have been reconstructed. We then compare our new data with earlier temperature proxy data from greenhouse intervals (Cenomanian-Turonian, Eocene) and to model simulations of these greenhouse periods. The caption of Figure 5 has been revised substantially and should be more explicit.

In their revision I would encourage the authors take a more logical, considered, and even skeptical approach. What if the shells are not as pristinely aragonite as their SEM and Raman data imply? The chalky and fractured nature of some of their samples from the photographs in Fig. 2 calls into question the ubiquity of their SEM and Raman-based conclusions. Similarly, is it possible that the burial temperatures are slightly warmer than the best estimates from the literature? Aragonite clumped isotope bond reordering is complex (relative to calcite and dolomite), poorly understood, and seemingly faster for a given thermal history than calcite (see Chen et al. 2019, GCA). The authors hardly dwell on this fact and its associated uncertainty.

As mentioned above, the section dedicated to the possibility clumped isotope reordering of may have been too quickly expedited and we will expand it substantially in the revised version to address these various aspects, including specificities related to aragonite mineralogy. We note however that the calcite and aragonite bivalve shells from NE France provided statistically indistinguishable clumped isotope values, in line with recent published data from the Jurassic of the UK (Vickers et al., 2021).

At the end of section 5.1, we have added a clear statement that we assume that the geochemical signal investigated here is very likely preserved, plus a consideration on the impact of a possible solid-state bond reordering on data interpretation.

Also, might the Polovinnaya River samples be estuarine, and not marine? Terrestrial fossils from the same shale exposures indicate that it might be a possibility, or at least one that needs detailed recognition even if it is not the preferred interpretation. An estuarine or non-marine origin might not impact the importance of their clumped isotope paleotemperatures, but it complicates the calculated water oxygen isotope ratios in ways that are interesting and not exclusive of comparisons in Fig. 5a.

This possibility is ruled out by the paleontological assemblages. Terrestrial fossils are entirely missing in the studied succession, except for fossil wood remains, which are common throughout shelf deposits and cannot be used as a proxy for marine / brackish / non-marine environments. All fossils recorded from the Polovinnaya section (bivalves, belemnites, forams) are fully marine. Mesohaline or brackish-water faunas are missing here. It should be noted that cephalopods are especially sensitive to salinity reduction, including belemnites (Hoffmann and Stevens, 2020). Although belemnites are sometimes considered as more tolerant to salinity decrease if compared with ammonites (Baraboshkin and Mutterlose, 2004), their occurrence is restricted to marine settings. The influx of fresh water and salinity decrease in early Toarcian of Siberia is possible, especially in those sites lacking ammonites. Saks and Nalnyaeva (1972) considered this issue during the discussion about overestimated isotope-based paleotemperatures derived from Toarcian belemnites of this area. An influence of freshwater influx during the early Toarcian was independently suggested by Kaplan (1976) in his studies of Mesozoic sedimentation of Siberia. Lastly, Protobranch bivalves (to which Dacryomya belongs) are not well adapted for salinities lower than 20‰ (Zardus, 2002). We will add some of these various and useful considerations to the revised version.

The discussion around salinity estimates for Polovinnaya (section 5.2, Line 320-342) has been reworked. We first present the biological constraints based on the fossil assemblage in comparison with our data. From a lower salinity limit based on the probable ecology of *Dacryomya*, we constrain a maximal $\delta^{18}O_p$ values of local runoffs. Using published estimates of polar $\delta^{18}O_p$ values during the Eocene warm period, we do the reverse exercise of estimating local salinity. All those salinity estimations are compared to results of climate simulations applied to the Toarcian.

Finally, in the text and in Figure 5 there is a casualness with comparing datasets over nearly 150 million years of the Mesozoic and Cenozoic, with dramatically different global paleoclimates and continental configurations, that makes the discussion hard to follow. For example, there is a large leap between the Early Jurassic and the Early Eocene on lines 254-259 that converts latitudinally ambiguous precipitation oxygen isotope ratios from Eocene proxy datasets to the calculation of paleosalinity during the Early Jurassic arctic. The leap is so large that it seems to obviate their point. In instances like this (see below for more line-specific commentary) I would encourage the authors to stick with data-model comparisons and well-reasoned hypotheticals. This reframing would still allow for the multi-period comparisons shown in Figure 5 with an edited discussion that better conforms to the study motivations outlined in the abstract and introduction.

We recognize that there is a large leap between these two time-intervals. Yet we believe our hypothesis are quite reasonable and explicitly stated, as we consider this approach as a better

alternative than simply using modern freshwater values, which would constitute an even larger leap of faith. Given the salinity tolerance of modern representative of the studied fossil (see above) an alternative fossil based approach would be to use a range of salinity to estimate freshwater isotope composition. Using 20-30 ‰ range for salinity and reconstructed $\delta^{18}O_{sw}$ values would give freshwater δ18O of ~-8‰ VSMOW for the lowest salinity hypothesis and down to ~-22‰ VSMOW for the highest salinity hypothesis.
We will add these complementary considerations to the revised version.

See the changes mentioned above regarding salinity calculations, which now are taken directly from the study site and then compared to Eocene values. As mentioned previously, section 5.3 was substantially rearranged to clarify the comparisons between our new data and published proxy-data and climate models, by first comparing temperatures then $\delta^{18}O_w$. We believe the revised version avoids large temporal leaps. We hope it clearer regarding these temporal aspects.

**Specific Comments**

**17-18** – The connection with the previous sentence is not very clear; why the distinction in time interval?

As explained in the introduction, there is no latitudinal gradient estimated for the Early Jurassic, as opposed to the Cretaceous-Early Paleogene periods.

**19** – "mildly buried" is a confusing term. Changed to 'shallow'

**24** – Correct "highlight" to "highlights". Not needed anymore as the correction suggested by Referee 1 was applied.

**36-38** – The authors could elaborate on this statement for better effect, I think. It may not be obvious to all readers how clumped isotopes are sensitive to burial.

We will add here references to better support these statements and develop this issue in the discussion for the revised manuscript.

We have added a phrase to explain these points and references to two papers modeling clumped isotope reordering, and one report of reordered data as an example.

**40** – "ante-Cretaceous" is an uncommon phrase. Changed to "predating the Cretaceous period"

**44** – Regarding "marine carbonate shells", there is some ambiguity on their marine origin in the discussion and I think the authors should specify that they are aragonitic fossils. This is important for two reasons: 1) aragonitic is exceptionally susceptible to geochemical alteration by conversion to calcite and 2) the bond reordering kinetics for aragonite are such that they are more prone to 'solid-state' clumped isotope change than calcite or dolomite.

We will add that the fossils presented here are mostly aragonitic (one bivalve shell from NE France is in calcite). We agree and are aware that aragonite is more prone to solid-state reordering. This will also be specified.

We have added "(mostly aragonite)" after "marine carbonate shells" to indicate the aragonite mineralogy of most specimens. Concerns regarding their marine origin is addressed in other comments.

**62 – 63** – "Exceptionally low burial" might be hyperbole. A Tmax of 420 °C in indeed immature, but not exceptionally so.
Removed "exceptionally"

**69-72 and 93-96** – It would be useful for the authors to commit to a maximum burial temperature. Using models of bond reordering and a simple burial history curve, it would be possible to estimate the possible change in clumped isotopes due to burial heating alone.

We will estimate possible changes in clumped isotopes using published models as suggested.

A maximal burial temperature is estimated for both studied sites based on estimated overburden and local geothermal gradient.

**Figure 2** – Samples (a), (b), (d), (i), (j), (k), and (l) all *look* too weathered or fractured to demonstrate that they retain primary shell material. Later it is revealed that (i) is not aragonitic (Fig. 3h). How are the authors able to admit that (i) is not aragonite, but call (e), (f), or (j) "pristine"? Each of these samples have the same coloration in these images.

It was not stated in the original manuscript and will be added it the revised version, but (e), (f) and (i) mostly show internal moulds after sampling of the shell with little aragonite material left. As for (j) most of the shell is lacking from mechanical alteration most probably while preparing and manipulating the specimen.
Indeed (i) is calcitic, but given the presence of mainly aragonite shells around it with no evidence for mineralogical conversion, it can be reasonably assumed the shell was originally in calcite.

We have added details on the preservation state of the specimens illustrated, especially when the specimen illustrated consists of the inner mould.

**102-103** – Regarding the white color, this chalky appearance can indicate shell alteration (mineralogical conversion or geochemically).

We must first mention that all samples are creamy white and not white as suggested here; perhaps our photographs do not do justice to their real aspect. Besides, we did not find any evidence for mineralogical conversion in either the Raman spectra or the SEM observations. We note that the superficial (optical) aspect of the shell can be dramatically altered by the mechanical crushing of

the shell, which is common in such fine-grained sediments and clearly visible in SEM images for some samples.

**Section 3.2.** – There are subscript and superscript formatting errors throughout this section. Corrected

**200** – Was the prismatic layer avoided when microsampling the shells?

This thin part evidences in the SEM images could not be avoided for technical reasons and bulk analysis of the shells were performed. We will add this aspect in the revised version.

**Line 122:** It is clearly stated that the shells were sampled as a whole. This phrase was already in the submitted version of July 2021. Here it has been moved from section 3.2 to section 3.1.

**Figure 3** – Why are the Raman spectra truncated <200 cm-1 for (b) and (d) and not for (f) and (h)? Also, it is not clear here (or in the text) how the position of these images and spectra relate to the subsamples for isotope analysis.

Different users took the spectra with slight different configuration. Yet this truncation does not hamper the identification of carbonate minerals. The main Raman shift rays used to differentiate calcite from aragonite are all above 200 cm-1 (282 and 713 cm-1 for calcite, and 209, 702 and 706 cm-1 for aragonite). The sampling strategy relative to the spectra (acquired on the sampled powders) and images will be clarified.

We have added this detail in the Method section (Line 122-130) relative to how the SEM images and Raman spectra were performed, to address how these data relate to the samples.

**249-252** – The temperature ranges cited are canonical values for calcite, not aragonite. We have added a phrase concerning aragonite susceptibility to solid-state bond reordering.

**261-263** – Regarding the possibility that small shell-water interaction has been shown to change clumped isotope ratios with only modest change in oxygen isotopes, what might be specific, independent evidence that this sort of thing had occurred (or not) in these shells?

Perhaps precise La-ICPMS or NanoSIMS could evidence isotopic gradients around fluid inclusion related to fluid inclusion-mineral interactions.

**263-265** – I think the implication here, subtly, is that 31 °C is something like a maximum burial temperature. Given that this is a fracture-fill carbonate without any other paragenetic sequencing context, it is equally possible that it's an exhumation temperature (i.e., a temperature experienced during fluid infiltration after maximum burial was reached).

This is correct, we do not have any data constraining the timing of this fracture infilling calcite. We will make it clear in the revised version.

We have added a phrase stating the uncertainty related to the formation date and depth of the considered vein.

**286** – I think it is notable that this range narrows considerably after removing the coolest temperature. The remaining 7 of 8 shells have an average of ~15 °C.

We will add the mean of our data to avoid confusion and better point the distribution of the clumped isotope temperature values.

We have added the mean of the data to better highlight data distribution.

**296-297** – Discussion of Jurassic food availability during the polar night, without any additional information, is entirely speculative and is too extrapolative from their isotope dataset.

We agree this section is speculative and it will be reduced to its minimum using supporting references.

The phrase was removed and this part now essentially refers to published work.

**302-303** – As mentioned above, this is an apples-to-oranges comparison. The similarity in SST between two shell populations separated in time by over a 100 million years might be entirely coincidental.

We found it reasonable to compare two datasets obtained using the same proxy on similar samples (bivalve shells) from high latitudes dated from periods both generally considered to register a warm climate, even though those periods are 140 million years apart.

The phrase was removed here. The salinity is now inferred from fossil assemblages to calculate oxygen isotope composition; the comparison of the Toarcian Eocene data has been moved as an opening at the end of the section (Section 5.3, line 385).

**330-342** – Also as mentioned above, I don't understand the relevance of Eocene high latitude precipitation values here. Who knows what they were in the early Jurassic?! As the authors show in Fig. 5, the modeled Jurassic poles were warmer than modeled Eocene poles, yielding lower latitudinal gradients in precipitation oxygen isotopes (see dashed lines for Eocene and Cretaceous data).

See the response above about the rationale used here. The referee might be confused here, as the modelled polar temperatures shown in figure 5 are actually much warmer in the most Eocene polar simulations than in the FOAM Early Jurassic simulation. Only the older 6x HadCM3 model yield similar temperatures. The reconstructed temperatures using proxy data are, however, quite similar (which is the reason why we used Eocene d18Op values).

We now state  that the Early Eocene was a "warm period with evidences of polar warmth", and refer to references that reported evidence of polar warmth for this period using different proxies. Those are the reason we consider the two periods to be comparable.
Maximum $\delta^{18}O_p$ values for the Early Jurassic Arctic are now first calculated using lower salinity limit provided by local fossil assemblage and then compared to those of the Eocene.

**348-349** – The authors should reframe this statement to consider an alternative scenario in which this locality and these shells are not marine at all. What if -4.9 to -2.5 are estuarine or mostly freshwater oxygen isotope values?

As stated above, the fossil data indicate marine conditions and rule out this possibility. In addition, assuming the analyzed interval was ice-free, such freshwater values would correspond to modern river values of –3.9 to -1.5 permil. This range of values in river water bodies is nowadays encountered only in warm tropical areas where precipitations are source from marine areas with very high rates of evaporation (e.g., Africa, central E Australia) and would imply climate conditions that are also at odds with available geological data. We will hence consider adding a phrase to rule out this possibility in the revised MS to make this point clearer to the readers.

Line 324 − We now state that the ecology of *Dacryomya* which set a lower range of 20‰ for local salinity.

**Figure 5** – Are the model results new or replotting of published results? If it's the latter than proper attribution needs to be clear in the figure or the figure caption.

The modelled temperatures and oxygen isotope values are replotted from published Earth system results. The associated references will be added in the revised manuscript.

We have added the meaning of the grey band, and the list of references used in the compilation of geochemical proxy data, as well as references associated with each climate simulation displayed to clarify this issue

**Overall modifications**

All references of newly cited documents were added accordingly.
Revision and homogenization of the formatting for numerical values and associated unit (including subscript for reference to isotope values).

New results from two additional aragonite specimens from Warcq were added. These new results are very close to previous ones, so they have no impact on the conclusions of the paper.
Newly published results from the literature were added to the compilation and are illustrated in Fig 5.